# Cerebral Infectious Opportunistic Lesions in a Patient with Acute Myeloid Leukaemia: The Challenge of Diagnosis and Clinical Management

**DOI:** 10.3390/antibiotics13050387

**Published:** 2024-04-24

**Authors:** Gabriele Cavazza, Cristina Motto, Caroline Regna-Gladin, Giovanna Travi, Elisa Di Gennaro, Francesco Peracchi, Bianca Monti, Nicolò Corti, Rosa Greco, Periana Minga, Marta Riva, Sara Rimoldi, Marta Vecchi, Carlotta Rogati, Davide Motta, Annamaria Pazzi, Chiara Vismara, Laura Bandiera, Fulvio Crippa, Valentina Mancini, Maria Sessa, Chiara Oltolini, Roberto Cairoli, Massimo Puoti

**Affiliations:** 1Department of Health Sciences, University of Milan Bicocca, 20126 Milan, Italy; gabriele.cavazza@ospedaleniguarda.it (G.C.); elisa.digennaro@ospedaleniguarda.it (E.D.G.); francesco.peracchi98@gmail.com (F.P.); bianca.monti@ospedaleniguarda.it (B.M.); nicolo.corti@ospedaleniguarda.it (N.C.); roberto.cairoli@ospedaleniguarda.it (R.C.); massimo.puoti@ospedaleniguarda.it (M.P.); 2Neurology and Stroke Unit, ASST Grande Ospedale Metropolitano Niguarda, 20162 Milan, Italy; cristina.motto@ospedaleniguarda.it (C.M.); maria.sessa@ospedaleniguarda.it (M.S.); 3Neuroradiology Unit, ASST Grande Ospedale Metropolitano Niguarda, 20162 Milan, Italy; caroline.regna-gladin@ospedaleniguarda.it; 4Infectious Diseases Unit, ASST Grande Ospedale Metropolitano Niguarda, 20162 Milan, Italy; giovanna.travi@ospedaleniguarda.it (G.T.); marta.vecchi@ospedaleniguarda.it (M.V.); carlotta.rogati@ospedaleniguarda.it (C.R.); davide.motta@ospedaleniguarda.it (D.M.); annamaria.pazzi@ospedaleniguarda.it (A.P.); fulvio.crippa@ospedaleniguarda.it (F.C.); 5Department of Haematology, ASST Grande Ospedale Metropolitano Niguarda, 20162 Milan, Italy; rosa.greco@ospedaleniguarda.it (R.G.); periana.minga@ospedaleniguarda.it (P.M.); marta.riva@ospedaleniguarda.it (M.R.); valentina.mancini@ospedaleniguarda.it (V.M.); 6Microbiology Unit, ASST Fatebenefratelli Sacco, 20157 Milan, Italy; sara.rimoldi@asst-fbf-sacco.it; 7Clinical Microbiology Unit, ASST Grande Ospedale Metropolitano Niguarda, 20162 Milan, Italy; chiara.vismara@ospedaleniguarda.it; 8Pathology Unit, ASST Grande Ospedale Metropolitano Niguarda, 20162 Milan, Italy; laura.bandiera@ospedaleniguarda.it

**Keywords:** cerebral abscesses, invasive fungal diseases, acute myeloid leukaemia, breakthrough central nervous system infection

## Abstract

Central nervous system (CNS) lesions, especially invasive fungal diseases (IFDs), in immunocompromised patients pose a great challenge in diagnosis and treatment. We report the case of a 48-year-old man with acute myeloid leukaemia and probable pulmonary aspergillosis, who developed hyposthenia of the left upper limb, after achieving leukaemia remission and while on voriconazole. Magnetic resonance imaging (MRI) showed oedematous CNS lesions with a haemorrhagic component in the right hemisphere with lepto-meningitis. After 2 weeks of antibiotics and amphotericin-B, brain biopsy revealed chronic inflammation with abscess and necrosis, while cultures were negative. Clinical recovery was attained, he was discharged on isavuconazole and allogeneic transplant was postponed, introducing azacitidine as a maintenance therapy. After initial improvement, MRI worsened; brain biopsy was repeated, showing similar histology; and 16S metagenomics sequencing analysis was positive (*Veilonella*, *Pseudomonas*). Despite 1 month of meropenem, MRI did not improve. The computer tomography and PET scan excluded extra-cranial infectious–inflammatory sites, and auto-immune genesis (sarcoidosis, histiocytosis, CNS vasculitis) was deemed unlikely due to the histological findings and unilateral lesions. We hypothesised possible IFD with peri-lesion inflammation and methyl-prednisolone was successfully introduced. Steroid tapering is ongoing and isavuconazole discontinuation is planned with close follow-up. In conclusion, the management of CNS complications in immunocompromised patients needs an interdisciplinary approach.

## 1. Introduction

Despite major improvements in the prophylaxis, diagnosis, and treatment of invasive fungal diseases (IFDs), they still represent an important cause of morbidity and mortality in patients affected by haematological malignancies (HMs), especially acute myeloid leukaemia (AML), and in allogeneic hematopoietic stem cell transplant (allo-HSCT) recipients. Recently updated guidelines by the consensus group of EORTC/MSG have refined the definitions of IFDs and breakthrough IFDs (b-IFDs), including advances in diagnostic tools [1,2]. In this setting of patients with HM, the diagnosis and management of central nervous system (CNS) disorders, including infectious complications, is a great challenge for clinicians. Herein, we report a case of a young patient with AML who developed CNS lesions while receiving effective antifungal treatment for probable pulmonary IFD. Despite many efforts to obtain a precise characterisation, the aetiological diagnosis of CNS lesions has remained presumptive to date with a negative impact on the effectiveness of therapies and a significant delay in oncological treatments.

## 2. Case Report

### 2.1. Leukaemia Diagnosis, Infectious Complications of Remission Induction and Consolidation Chemo-Therapies, Diagnosis and Management of Central Nervous System Lesions

#### 2.1.1. Case Presentation

A 48-year-old male patient was diagnosed with AML with intermediate ELN 2022 risk with cryptic translocation t(9;11)(p21;q23) involving KMT2A gene and chromosome 8 trisomy. At the onset of HM, he received medium–high doses of steroids for haemophagocytosis. Concomitantly to remission-induction chemotherapy with idarubicin and cytarabine (3 + 7), he developed probable endo-bronchial aspergillosis. Chest computed tomography (CT) showed interstitial lesions suggestive of haemorrhagic alveolitis (Figure 1A). The galactomannan antigen level in bronco-alveolar lavage fluid (BAL-f) tested with a chemiluminescent immunoassay (CLIA) was 8.7 and BAL-f culture was positive for triazoles-susceptible *Aspergillus flavus*. The patient received antifungal therapy (AFT) with liposomal amphotericin-B (L-AMB) and he achieved complete morphological remission of AML. Upon hospital discharge, AFT was switched to voriconazole.

Two months later, the patient underwent consolidation chemotherapy with high-dose cytarabine and idarubicin, attaining both morphological and cytogenetic complete remission. Before starting consolidation chemotherapy, a chest computed tomography (CT) demonstrated thickening/atelectasis of the middle lobe (Figure 1B), so a bronchoscopy was repeated with evidence of whitish plaques on the right bronchial hemi-system. Needle aspiration cytology was non-conclusive, galactomannan antigen on BAL-f was 2.7, and BAL-f culture showed growth of *Aspergillus fumigatus*; the AFT with voriconazole was continued with dose adjusted according to therapeutic drug monitoring (TDM) (range of basal TDM: ≥1 mg/L) in the absence of toxicities. After one month, the patient underwent a second cycle of consolidation chemotherapy; at that point, the chest CT scan showed a reduction in the atelectasis of the middle lobe and the appearance of atelectasis of the right inferior lobe (Figure 1C). Another bronchoscopy was performed, revealing a sub-obstruction in the lumen of the right main bronchus consistent with localizations of endo-bronchial aspergillosis, despite BAL-f culture being negative and galactomannan antigen being 1.4, warranting a continuation of voriconazole. One month later, the patient went to the Emergency Department because of hyposthenia of the left upper limb. Brain magnetic resonance imaging (MRI) showed oedematous lesions with a haemorrhagic component and ring contrast enhancement in the cortical–subcortical frontal, parietal, and occipital right hemispheres with lepto-meningitis (Figure 2A).

At admission, empirical antimicrobial therapy was initiated with meropenem 2 g every 8 h, linezolid 600 mg every 12 h and L-AMB 5 mg/kg daily. The lumbar puncture was performed for cerebral spinal fluid (CSF) analysis three days after admission, revealing no white blood cells (<5 cells/µL), normal glucose levels (50 mg/dL, glucose serum/CSF ratio 0.37), mild hyper-protidorrhachia (proteins level 75 mg/dL), no bacteria or filamentous fungi on direct staining nor in cultures, negative galactomannan antigen (0.068), and negative *Cryptococcus* antigen. Moreover, the lung CT scan excluded aspergillosis relapse (Figure 1D). Ten days after admission, brain MRI worsened, with progression of peri-lesion oedema in right hemispheric white matter and the size of enhancing lesions (Figure 2B). So, a steroid was introduced (dexamethasone 8 mg every 12 h for ten days), obtaining hemi-syndrome resolution. Two weeks after admission and five days after steroid initiation, a brain biopsy was performed. The histology revealed a chronic inflammatory process with abscesses and necrosis, Grocott’s staining was negative for spores and hyphae, bradyzoites of *Toxoplasma* spp. were not seen, and myeloid sarcoma was excluded; finally, bacterial (including slow-growing pathogens), fungal, and mycobacterial cultures were negative. One week after the brain biopsy, the antimicrobial therapy was empirically modified to trimethoprim/sulfamethoxazole, discontinuing meropenem and linezolid after three weeks of treatment. Two weeks later, brain MRI showed an improvement in both oedema extension and the size of enhancing lesions (Figure 2C). After five weeks of L-AMB, AFT was modified to isavuconazole 200 mg daily after the standard loading dose and antimicrobial treatment were empirically modified to linezolid 600 mg every 12 h and moxifloxacin 400 mg daily due to myelotoxicity after three weeks of trimethoprim/sulfamethoxazole therapy. The patient was discharged home in good condition with complete clinical recovery. Regarding the oncological treatment of AML, despite initial plans for allogeneic hematopoietic stem cell transplant (allo-HSCT), this was postponed due to CNS complications, with oral azacitidine introduced as a maintenance therapy for AML.

#### 2.1.2. Case Discussion and Literature Review

The main recommendations regarding the diagnosis of brain abscesses and opportunistic CNS lesions in the ESCMID guidelines advocate for MRI as the gold-standard imaging method for diagnosis. In patients without severe presentation, the withdrawal of antimicrobials until aspiration or excision of the brain abscess if neurosurgery is feasible and can be carried out within a reasonable time (preferable within 24 h) [3] is advised. In a meta-analysis conducted by the authors, withholding antimicrobials until neurosurgical excision led to a significantly higher proportion of patients achieving a microbiological diagnosis compared to those who received antimicrobials before excision [123/153 (80%) vs. 41/126 (33%)] [3]. For empirical treatment of community-acquired brain abscess, the guidelines recommend an antibiotic therapy with third-generation cephalosporin and metronidazole. Additionally, trimethoprim/sulfamethoxazole and voriconazole may be added in immunocompromised hosts. Glucocorticoid therapy is suggested as an adjunctive treatment for severe symptoms due to peri-focal oedema or impending herniation [3]. Our case highlights two critical aspects in the diagnosis: firstly, the timing of brain biopsy was significantly delayed (after two weeks of broad-spectrum antimicrobials), thus reducing the sensitivity of the cultures; moreover, the initiation of steroids before the biopsy could have partially influenced the results as a confounding factor regarding the differential diagnosis with CNS vasculitis. Secondly, the availability of molecular-based techniques (both the detection of *Aspergillus* PCR in CSF and the molecular detection of bacteria and fungi in biopsy samples) is not widespread or easily attainable in the daily practice. In our case, these aspects translate into a low negative predictive value of the microbiological cultures of the brain biopsy. Analysing the neuro-imaging characteristics of our patient, the presence of meningeal enhancement and brain abscesses with haemorrhagic components could suggest the diagnosis of possible invasive aspergillosis (IA) of CNS through haematogenous dissemination [4]. However, the negative galactomannan antigen in serum and CSF, whose sensitivity is known to be reduced by mould-active azoles (i.e., voriconazole in our patient), and the unavailability of *Aspergillus* PCR did not allow the patient to meet the mycological criterion to define the diagnosis of probable SNC IA [1]. The microscopic examination of the brain biopsy did not demonstrate tissue invasion by hyphae; however, the sensitivity of microscopy for IA is 50% at best, as reported in the ESCMID guidelines on *Aspergillus* diseases [4].

IFDs of the CNS are life-threatening infections in patients with HM; however, analyses of large cohorts of patients focusing on these rare IFDs are still lacking. One of the largest is from SEIFEM, which analysed 89 consecutive cases of proven (*n* = 53) or probable (n = 36) CNS IFDs (71/89) and para-nasal sinus IFDs (18/89). The most common underlying disease was acute leukaemia (69%) and 29% of patients previously underwent allo-HSCT; additionally, IFDs occurred in particular at leukaemia onset or in relapsed/refractory diseases, with only 19% in the setting of consolidation chemotherapy. *Aspergillus* spp. was the most common pathogen (69%), followed by mucormycetes (22%), *Cryptococcus* spp. (4%), and *Fusarium* spp. (2%), and in 48% of cases the lung was the primary site of the IFD. Among the CNS IFDs, CNS biopsy and an analysis of galactomannan antigen in CSF were performed in 10% and 42% of cases, respectively, and the presence of galactomannan antigen in CSF was positive in 67%. Although most patients received ≥2 lines of therapy (58%) and almost half were treated with a combination of ≥2 drugs (38%), the overall response rate to AFT (complete or partial) was 57%, and the 1-year overall survival was poor (32%) with an IFD-attributable mortality of 33% [5]. Another series of 40 patients with HM and CNS invasive mould diseases (IMDs) (n = 16 proven IMDs) was reported by the MD Anderson Cancer Center. The incidence density was 3.8 cases/100,000 patient days and most patients had active HM and neutropenia at diagnosis (95% and 53%, respectively). Among cases with a microbiological diagnosis (25/40), most were represented by *Aspergillus* spp. and *Mucorales* (85%). CNS IMDs were deemed to be secondary to hematogenous spread in 31/40 (77%), and were mostly fungal pneumonia (28/31, 90%). CNS IMDs was diagnosed concomitantly with fungal pneumonia in 3/28 (11%); in the remaining 25/28 cases (89%), CNS IMDs occurred at a median of 15 days (range: 5–283 days) after the diagnosis of fungal pneumonia. The galactomannan antigen was positive in 33% of patients who had CSF tested (3/9). CNS lesions typically presented as solitary ring-enhancing abscesses in MRI (26/40), while vascular events with a stroke-like picture occurred in nine patients. Most of the patients (85%) received L-AMB and were treated with a combination AFT (83%). The mortality at 42 days was 48%; in the univariate and multivariable analyses, improved survival was related to immune response in histopathology (presence of giant cells and granulomas), the absence of co-infections, corticosteroid tapering, and possibly surgical drainage [6]. Considering the present case, our patient developed CNS opportunistic lesions after achieving complete remission of AML and approximately four months after the diagnosis of probable endo-bronchial aspergillosis without the evidence of relapsed/refractory pulmonary IFD. This diminishes confidence in the diagnosis of possible breakthrough CNS IA. However, as recommended by the guidelines [4], AFT was modified to a different drug class, switching from voriconazole to L-AMB. After the course of antifungal therapy with L-AMB, we decided to switch to isavuconazole for several reasons. Firstly, we were not confident in resuming the same drug (voriconazole) on which the CNS complication occurred; at the same time, a simplification to an oral antifungal was deemed reasonable. Isavuconazole was preferred because of previous reports that suggested a good clinical response in patients with CNS IFDs (36 patients; complete or partial clinical response at the end of treatment: 58%; survival at day 42: 80%) [7] and its potential to achieve efficacious concentrations in rat brains comparable to those of voriconazole [8]. Conversely, data on posaconazole CNS penetration and its efficacy in the treatment of CNS IFDs seem more variable with CSF/plasma concentration ratios of 0.009 [9,10].

Given the absence of a definitive diagnosis of proven–probable CNS IFD or neurotoxoplasmosis, we maintained antibiotic therapy with linezolid and moxifloxacin, targeting possible *Nocardia* spp. and *Actinomyces* spp. As recently reported by the Infectious Disease Working Party of European society of Blood and Marrow Transplant that described 81 cases (74/81 allo-HSCT recipients), nocardiosis is a common cause of CNS infections in HM patients, particularly allo-HSCT recipients. Nocardiosis was a late-onset complication occurring a median of 8 months after HSCT and the brain was involved in 37% of cases (30/81) with multiple brain abscesses (19/30, 63%). *Nocardia farcinica* was the most common identified species and the highest susceptibility rates were reported for linezolid (100%), trimethoprim/sulfamethoxazole (90%), and imipenem (86%) [11]. A recent Israeli study analysed the susceptibility of 138 clinical strains of *Nocardia* spp. (broth micro-dilution for MIC value assessment, and whole-genome sequencing for species identification and antimicrobial resistance gene and mutation analysis), finding that linezolid was active against all isolates (100%), followed by trimethoprim/sulfamethoxazole (93%) and amikacin (91%), while resistance to other antibiotics was species-specific [12]. Linezolid also displayed good coverage against *Actinomyces* spp., as reported by a study that analysed the antimicrobial susceptibility of 100 clinical oral isolates. No to low resistance (0–2%) was observed against penicillin, ampicillin/sulbactam, meropenem, clindamycin, linezolid, and tigecycline, while a high level of resistance to moxifloxacin occurred (>80%) [13]. Neuro-toxoplasmosis was also considered in the differential diagnosis, even though in our patient bradyzoites of *Toxoplasma* were not seen in the histological samples of brain biopsy and his serological test for IgG antibodies was negative. Even if severe toxoplasmosis is more common in HM patients after allo-HSCT, patients with chronic lympho-proliferative disorders or acute leukaemia can also develop this opportunistic infection. A review of 44 cases reveals that toxoplasmosis occurs, mainly, in patients with lympho-proliferative disorders and rarely in patients with acute leukaemia; the CNS is the main organ involved, the absence of chemoprophylaxis is a risk factor for opportunistic toxoplasmosis, and the global mortality is >50% [14]. Finally, another infectious cause of CNS infections in patients with AML is *Bacillus cereus*, a ubiquitous Gram-positive rod-shaped bacterium that can cause sepsis and neuroinvasive disease in patients with AML or neutropenia. Recently, a case series on this uncommon opportunistic CNS infection was reported by Brigham and Women’s Hospital and the Dana-Farber Cancer Institute. Five cases of neuroinvasive *Bacillus cereus* disease were described; all cases were secondary to prolonged and severe neutropenia in patients with AML during induction chemotherapy and an environmental source was excluded via sequencing. Neurologic involvement included subarachnoid or intra-parenchymal haemorrhage or brain abscesses and all patients received ciprofloxacin and survived [15].

### 2.2. Clinical and Neuroimaging Follow-Up of Central Nervous System Infectious Lesions

#### 2.2.1. Case Presentation

One month after hospital discharge, brain MRI showed a minimal increase in lesions. Consequently, azacitidine was discontinued and another MRI a month later showed further slight worsening (Figure 2D). At that point, while maintaining the same antimicrobial treatment (linezolid, moxifloxacin, isavuconazole), the patient underwent another brain biopsy, excising the nodular lesion at the occipital–parietal right lobe. After the surgical procedure, in the absence of major complications, the patient was discharged home continuing only AFT with isavuconazole, while waiting for histological and microbiological results. The histology was the same as the previous biopsy. It showed brain parenchyma with acute and chronic inflammatory processes surrounded by a fibrous capsule indicative of an abscess in the organising phase (mycelial hyphae, alcohol acid-resistant bacilli, and bradyzoites of *Toxoplasma* were not observed). Once again, the cultures resulted negative (standard and slow-growing bacteria, mycobacteria, filamentous fungi) and the bacteria sequencing through NGS 16S metagenome analysis (Ion Torrent, Thermofisher, Waltham, MA USA) was positive for *Veilonella* spp. and *Pseudomonas* spp (read frequency 8000 for both pathogens). Based on those results, one month after the brain biopsy, the patient was again hospitalised to receive intravenous antibiotic treatment with meropenem 2 g every 8 h in extended infusion while continuing isavuconazole. At in-hospital admission, the patient was asymptomatic. A brain MRI was repeated, showing post-operative changes in the right parietal region and a further slight increase in the remaining cortico–pial enhancement and oedema in the right hemisphere; no new lesions or abnormalities in different vascular territories were described (Figure 2E). The TDM was assessed to attain plasma level ≥ 20 mg/L for meropenem with dose adjustment (8 g daily in continuous infusion) and ≥3 mg/L for isavuconazole without dose adjustment (200 mg daily). A dental CT scan excluded periodontitis as a possible predisposing factor for the development of brain abscesses. After one month of carbapenem therapy and without the onset of any clinical signs or symptoms, brain MRI documented a progressive increase in contrast-enhancement nodular lesion size and oedema extension in the right parietal and frontal cortico–pial areas, besides the surgical cavity. Thus, meropenem was discontinued and the differential diagnosis was re-discussed.

The patient underwent a total-body CT scan and positron emission tomography, which ruled out extra-cranial infectious–inflammatory localisations, with resolution of the endo-bronchial aspergillosis (Figure 1E) that was confirmed by bronchoscopy with negative galactomannan antigen; negative bacterial, fungal, and mycobacterial BAL-f cultures; and negative cytology (Grocott and Ziehl–Neelsen staining). Additionally, para-sinus involvement was excluded. After multidisciplinary discussion, the auto-immune genesis (sarcoidosis, histiocytosis, or primary CNS vasculitis) was deemed unlikely due to histological findings (not diagnostic for myeloid sarcoma, primary CNS vasculitis, sarcoidosis, or histiocytosis), the absence of sinus involvement (which is usually present in the case of histiocytosis), the pauci-symptomatic clinical course, and the negativity of the auto-immune panel (anti-extractable nuclear antigens, anti-nuclear antibodies, anti-neutrophil cytoplasmic antibody, anti-dsDNA, anti-centromere, rheumatoid factor, and angiotensin-converting enzyme). Cerebral CT angiography showed no large or medium vessel stenoses suggestive of primary CNS vasculitis. A digital subtraction cerebral angiography was not performed. The bone marrow evaluation confirmed complete remission of AML. Furthermore, although no fungal hyphae were detected in the histology of the brain biopsies, thus reducing the negative predictive value of the test, it was decided to perform fungal metagenomics (18S and 28S r-RNA gene sequencing) on the histological material of the second brain biopsy and the result was negative. Therefore, the leading hypothesis was CNS opportunistic lesions, namely, possible CNS IFD as the most conceivable aetiology, with a peri-lesion inflammatory response. As such, steroid therapy was started with dexamethasone 8 mg every 12 h (methyl-prednisolone equivalent dose of 1 mg/kg daily) while continuing isavuconazole. Two weeks later, brain MRI improved, in both lesion size and oedema (Figure 2F). Steroid tapering was initiated and, three weeks later, another brain MRI was performed, showing further reductions in oedema.

After multidisciplinary discussion and considering the long-term clinical and neuroradiological follow-up 9 months post onset of CNS lesions, along with the above-mentioned microbiological and histological results and response to antimicrobial treatments and steroid therapy, it was decided not to further extend the immunosuppressive therapy with dexamethasone. The most likely aetiology was considered opportunistic fungal, with a subsequent peri-lesion inflammatory response rather than a secondary para-infectious vasculitis. To date, the steroid has been stopped and, concomitantly, a brain MRI was performed 6 weeks after the previous one, revealing a further improvement in oedema and unchanged CNS lesions. AFT with isavuconazole has been discontinued and a close clinical and neuroradiological follow-up has been planned to detect possible worsening of CNS lesions early and repeat the diagnostic work-up in a timely manner (lumbar puncture, brain biopsy). The patient currently maintains complete molecular and morphological remission of AML; however, allo-HSCT was not performed because of the CNS complication, and, for this reason, he retains a greater probability of AML relapse.

#### 2.2.2. Case Discussion and Literature Review

In the diagnosis of brain abscesses, ESCMID guidelines recommend the use of molecular-based diagnostics in cases of negative cultures to increase the diagnostic yield of brain abscess material and to guide the choice of antimicrobials at early and later stages (e.g., treatment failure, as in our case) [3]. A total of nine studies (five prospective, three multicentre) were included in the analysis that led to the conditional recommendation. The pooled analysis showed that molecular diagnostics were concordant-positive with the culture results in 187/280 (67%), concordant-negative in 22/280 (8%), only culture-positive in 24/280 (9%), only molecular-diagnostics-positive in 36/280 (13%), and discordant-positive in 13/280 (5%). Molecular diagnostics expanded the number of identified pathogens (especially anaerobic bacteria) in 115/173 (66%) cases of brain abscess caused by oral cavity bacteria compared with culture [3]. The potential role of molecular-based diagnostics in the specific setting of patients with HM that received allo-HSCT and developed CNS complications was evaluated in a Chinese series of 20 patients whose CSF was tested with a metagenomics next-generation sequencing (m-NGS) technique. In five patients, the m-NGS was negative, and an alternative non-infectious diagnosis emerged (n = 3 posterior reversible encephalopathy syndrome, n = 1 transplant-associated thrombotic micro-angiopathy, n = 1 transient seizure). In the remaining fifteen patients, the m-NGS technique was positive, frequently for more than one pathogen, and highly sensitive, posing a challenge in result interpretation. The m-NGS analysis documented mainly viruses (10 CMV, 3 BKPyV), then bacteria (coagulase-negative staphylococci and *E. faecium*, non-fermenting Gram-negative bacteria, and no bacteria were isolated by standard cultures) and less frequently fungi (*Aspergillus* spp., *Fusarium* spp., *Penicillium* spp.) [16].

In our case, 16S metagenome sequencing revealed the presence of both *Veilonella* spp. and *Pseudomonas* spp.; consequently, the clinical value of those results was discussed. Given the sub-acute and almost asymptomatic course, brain lesions caused by *Pseudomonas* spp. were considered unlikely, while the potential role of *Veilonella* spp., often missed by cultures and well known as a cause of brain abscess, was noted [3]. The isolation of anaerobic bacteria usually indicates poly-microbial infections and aerobic bacteria are often not cultured because an effective antibiotic therapy exists. For this reason, we decided for a carbapenem-based treatment with an active spectrum against both aerobic and anaerobic bacteria, including *Pseudomonas aeruginosa*. Susceptibility data are relatively scarce on anaerobic organisms. Recently, a Canadian study determined the antimicrobial susceptibility profiles for more than 5000 clinically significant anaerobic bacteria (17% *Bacteroides* spp., 14% *Clostridium* spp., 12% *Cutibacterium* spp., 7% *Actinomyces* spp., 1% *Veilonella* spp.) using the gradient strip method (MIC value interpretation according to CLSI guidelines). The seventy-three *Veilonella* spp. isolates displayed an excellent susceptibility to meropenem (97%), metronidazole (100%), and clindamycin (96%) [17]. Another study evaluated the susceptibility to antibiotics (ampicillin, piperacillin/tazobactam, cefoxitin, tetracycline, moxifloxacin, clindamycin, metronidazole, and vancomycin) of four species of *Veillonella* spp. With the agar dilution method, showing that all species were susceptible to cefoxitin, tetracycline, moxifloxacin, clindamycin, and metronidazole, while vancomycin susceptibility varied greatly according to species [18]. It was decided to use meropenem, considering that a sub-optimal therapy against *Veilonella* spp. was previously administered because linezolid’s spectrum is mainly active against Gram-positive bacteria (including anaerobic) and moxifloxacin; even if it is potentially active against *Veilonella* spp., it may have resulted in a sub-optimal dose exposure in the difficult-to-treat CNS site of infection. However, despite the TDM assessment for dose adjustments to attain adequate plasmatic levels, carbapenem therapy resulted in treatment failure.

At that point, differential diagnoses were re-discussed with neurologists, neuro-radiologists, haematologists, and immunologists. Different aetiologies of aseptic meningitis were evaluated, including systemic diseases with meningeal involvement (sarcoidosis, Behçet disease, Sjögren syndrome, systemic lupus erythematosus, granulomatosis with poly-angiitis), neoplastic meningitis, and infectious diseases, mainly viral and fungal. Infections are a frequent cause of cerebral vasculitis, which can be caused by angiotropic pathogens (varicella zoster virus, *Treponema pallidum*, *Aspergillus*) [19]. Some reports highlighted the association between CNS IA and secondary vasculitis of large vessels, usually documented by histology, and the need for steroid therapy alongside AFT [20,21,22]. Although the histology of the brain biopsy in our case did not support the presence of vasculitis and the cerebral CT angiography was unremarkable (despite its low sensitivity for vasculitis), considering the increasing peri-lesion oedema of CNS lesions at brain MRI without the appearance of new localisations during the follow-up, a course of steroids (similarly to the gluco-corticoid regimen adopted for remission induction in non-severe forms of giant cell arteritis [23]) was administered. This therapeutic measure achieved partial neuro-radiological response, but the clinical picture did not lead us to a further continuation of immunosuppressive therapy. To date, despite many efforts to obtain a precise characterisation, the aetiological diagnosis of CNS lesions in the patient has remained presumptive, negatively impacting the effectiveness of therapies and the management of AML. CNS complications represent a diagnostic and clinical challenge in HM patients, particularly in allo-HSCT recipients, where infections account for approximately 30% of CNS complications, primarily due to IFD, toxoplasmosis, and herpes virus encephalitis. The overall mortality rate, due to CNS complications, is considerable, ranging from 20% to 25%, with infections being the leading cause of mortality related to CNS complications [24].

## 3. Conclusions

This case report perfectly exemplifies the challenges in diagnosing, managing, and treating infectious opportunistic CNS lesions in a patient with AML. The primary key messages from our clinical case can be summarised as follows: (i) in immunocompromised hosts with CNS lesions, diagnostic procedures, such as lumbar puncture or brain biopsy, are of paramount importance because attaining a correct characterization is fundamental for patients’ treatment and management; (ii) their timely execution should be pursued to maximise the microbiological yield and to not delay antimicrobial therapies too long; and (iii) the use of novel molecular-based techniques to attain a microbiological diagnosis is a diagnostic tool to be implemented in daily practice, while considering the need for a careful interpretation according to the patient’s setting and the clinical course. Despite many efforts in the diagnostics and availability of novel antimicrobial drugs, the aetiological diagnosis of brain lesions, including brain abscesses, is frequently missed, leading to critical patient outcomes, both for infection-related mortality and the delay in haematological treatments, resulting in a higher risk of AML non-remission or relapse. Additionally, in cases of CNS invasive fungal diseases (IFDs), particularly in cases of Aspergillus-associated vasculitis, clinicians should consider the potential role of steroids. In conclusion, managing immunocompromised hosts with CNS lesions, potentially due to opportunistic infections, necessitates a multidisciplinary approach to achieve a correct diagnosis, optimise therapy appropriateness, perform differential diagnosis, and enhance patient management.

## Figures and Tables

**Figure 1 antibiotics-13-00387-f001:**
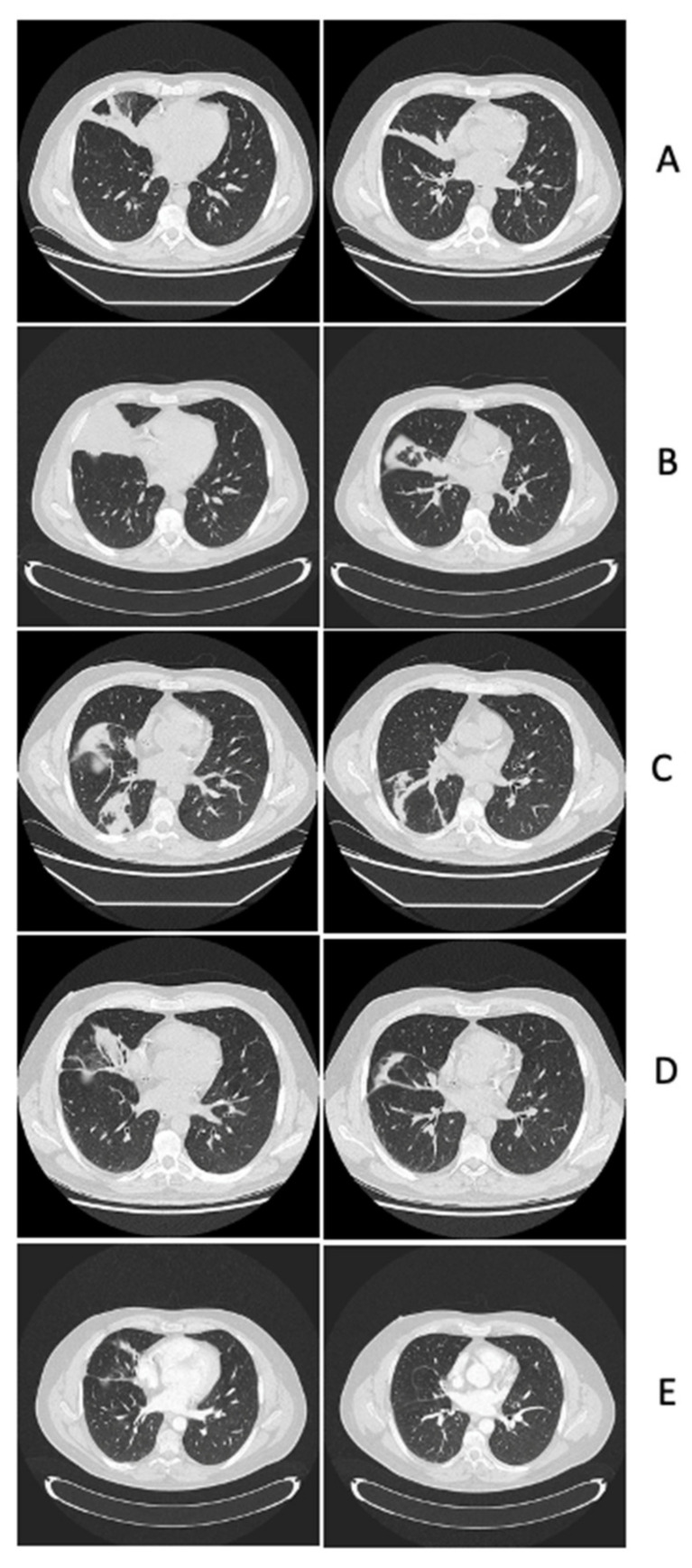
This is the timeline of lung computer tomography scans: (**A**) Leukaemia diagnosis, initiation of remission induction chemotherapy: interstitial pneumonia with haemorrhagic alveolitis in the medium lobe. (**B**) Before first cycle of consolidation chemotherapy: complete atelectasis of the medium lobe with obliteration of the bronchus. (**C**) Before second cycle of consolidation chemotherapy: reduction in the atelectasis of the medium lobe and appearance of atelectasis of the right inferior lobe sustained by the sub-obstruction of the lumen of the right main and intermediate bronchi. (**D**) At the onset of opportunistic central nervous system lesions: reduction in atelectasis of both the medium and right inferior lobes. (**E**) Before starting corticosteroid treatment for central nervous system lesions: resolution of endo-bronchial/pulmonary aspergillosis.

**Figure 2 antibiotics-13-00387-f002:**
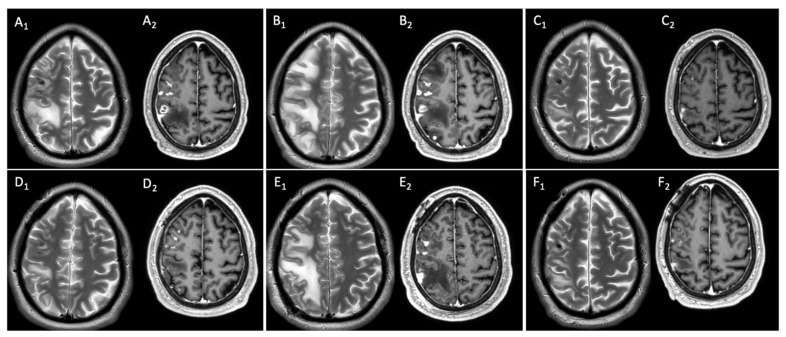
This is the timeline of brain magnetic resonance imaging: (**A_1_**) Axial T2-TSE w.i.: multiple hyperintense subcortical foci of oedema with mass effect in right fronto-parietal region. (**A_2_**) Post-contrast FFE-T1 w.i.: cortico-subcortical frontal and parietal foci of inhomogeneous enhancement, necrosis, and meningeal involvement. (**B_1_**) Axial T2-TSE w.i.: worsening of oedema and mass effect. (**B_2_**) Post-contrast FFE-T1 w.i.: increase in size of frontal and parietal foci of enhancement. (**C_1_**) Axial T2-TSE w.i.: reduction in oedema and mass effect. (**C_2_**) Post-contrast FFE-T1 w.i.: signs of right frontal craniotomy and remarkable reduction in enhancing parietal lesion. (**D_1_**) Axial T2-TSE w.i.: increase in oedema extension. (**D_2_**) Post-contrast FFE-11 w.i.: increase in size of enhancing frontal and parietal lesions. (**E_1_**) Axial T2-TSE w.i.: further progressive extension of oedema in right hemisphere subcortical white matter. (**E_2_**) Post-contrast FFE-T1 w.i.: progressive increase in enhancing frontal and parietal lesions. (**F_1_**) Axial T2-TSE w.i.: partial reabsorption of oedema in right hemisphere and reduction in mass effect. (**F_2_**) Post-contrast FFE-T1 w.i.: decrease in enhancement in both frontal and parietal regions.

## Data Availability

Data are available from the corresponding author upon reasonable request.

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
