# Peer review of "Cerebral Infectious Opportunistic Lesions in a Patient with Acute Myeloid Leukaemia: The Challenge of Diagnosis and Clinical Management"

_antibiotics, 2024, doi:10.3390/antibiotics13050387_

Round 1

Reviewer 1 Report

Comments and Suggestions for Authors

This is a paper describing a case of acute myeloid leukaemia with underlying probable endobronchial Aspergillosis, subsequently developing into a brain abscess. The patient developed clinical deterioration despite broad-spectrum antibiotics and anti-fungal coverage, with subsequent clinical picture compatible with immune reconstitution, therefore requiring steroid use. The case illustrated the difficulty in obtaining a microbiological diagnosis of brain abscess, and the need for empirical treatment when diagnosis is uncertain. 

Overall comments: 

This case illustrates the common difficulty in managing opportunistic infections in immunocompromised hosts. The case is clearly written, however, some of the clinical judgement should be justified or further elaborated in this case (including why sequencing was not done on the initial specimen to ascertain diagnosis, the choice of anti-fungal agents etc.). There is extensive use of abbreviations in this paper, the full form of some of the abbreviations should be mentioned, and there are occasions where the abbreviations were wrongly written in the paper. The paper would benefit from further English editing. The detailed comments are listed below for the authors’ reference. The bold comments represent comments that need further input.

Specific comments:

Abstract: 

- Line 22: “infectious-opportunistic” is not necessary. Please delete

- Line 28: Do you mean 16S rRNA or mRNA?

Introduction:

- Line 48: “are a great challenge” - is a great challenge or are great challenges

- Line 48-49: “We report and the case” - delete “and the”, replace it with “a”

Case report:

- Line 76: Delete “scan”

- Line 79: Any susceptibility testing performed on the strain of Aspergillus fumigatus?

- Line 93: Delete “in-hospital” - admission already has the meaning of “in-hospital”

- Line 105-108: This sentence should be rewritten, including phrases such as “bacterial, fungal, and mycobacterial culture were negative”, “bradyzoites of Toxoplasma were not seen”

- Line 112-113: Please discuss and justify why isavuconazole but not other azoles were used in this stage. Should note that some studies have shown that isavuconazole has poor CNS penetration when compared with other azoles such as posaconazole and voriconazole. Are there any reasons to support the use of isavuconazole in this patient?

- Line 117-119: Please rewrite this sentence, as the meaning is uncertain.

- Line 138-139: “Moreover, the initiation of steroids before the biopsy could have partially influenced the results as a confounding factor.” - How do steroids affect the microbiological culture of brain biopsy? 

Line 150, 187, 271, 348 (and other occasions): Should be CNS instead of SNC

- Line 163: Please change that statement to “Galactomann antigen on CSF was positive in 67% of cases.”

- Line 192-193: Please rewrite this sentence, as the meaning is uncertain.

- Line 194: The spelling of Actinomyces spp. is incorrect.

- Line 211-214: Please rewrite this sentence, as the meaning is uncertain.

- Line 216: “Can developed” - change to “can develop”

- Line 220-226: Please elaborate on the source of Bacillus cereus in the included study, and emphasise that it is neutropenia that leads to Bacillus bacteraemia.

- Line 243: Please rewrite the sentence in the bracket. We almost do not see spores in patients with invasive fungal infection, spores may be seen in cases of Coccidioides immitis, therefore I think spores should be mentioned here when we are talking about histology of tissue. Please use the word seen or observed instead of retrieved. Toxoplasma should be italicised in this sentence.

- Line 246: Please elaborate on how your laboratory does 16S rRNA sequencing. Is it performed on fresh specimens? Is it performed using Sanger sequencing or NGS? If Sanger, then the result is strange as usually only one organism can be detected using this methodology. If NGS, you need to provide further information in terms of the results generated for respective organisms, including the number of reads, the NGS platform, etc.

- Line 254: “plasmatic” - should be changed to plasma

- Line 266: “standard bacteria, filamentous fungi, slow-growing/demanding bacteria” - please change to bacterial, fungal, and mycobacterial culture.

- Line 273-274: Please elaborate on the full form of all abbreviations when mentioning these autoimmune markers, as not all readers understand the meaning of the abbreviations.

- Line 280-281: One of the reasons for a negative 18s/28s sequencing is that the sequencing was performed on a specimen with previous exposure to anti-fungal. Was fungal sequencing considered in the first specimen saved for this patient? The yield of sequencing should be higher in that case. 

- Line 328-333: I agree with some of the interpretations by the authors, however, the authors should note that the isolation of anaerobic bacteria represents polymicrobial infection and only anaerobic bacteria detected may mean that the majority of aerobic bacteria were covered by previous antibiotics. This point should be emphasised in the discussion section.

- Line 358: “Syphilis” - should be changed to “Treponema pallidum” as you are referring to bacteria instead of the disease entity 

- Line 363: sensibility - should be changed to sensitivity

Figures:

- Line 389 - 390: “Timeline of brain MRI” should be deleted.

Comments on the Quality of English Language

As above

Author Response

Abstract: Line 22: we have deleted “infectious-opportunistic”. Line 28: We mean 16S rRNA, we have modified accordingly.

Introduction: Line 48: we have replaced with “is a great challenge”. Line 48-49: We have corrected the sentence.

Case report: Line 76: we have deleted  “scan”. Line 79: unfortunately, the susceptibility test was not performed on the strain of Aspergillus fumigatus. Line 93: we have deleted the term “in-hospital”. Line 105-108: we have re-written the sentence as suggested by the Reviewer. Line 112-113: we have addressed the discussion of our therapeutic choice in the section “2.1.2 Case discussion and literature review”. Line 117-119: we have revised the sentence to better clarify its meaning. Line 138-139: “Moreover, the initiation of steroids before the biopsy could have partially influenced the results as a confounding factor.” How do steroids affect the microbiological culture of brain biopsy? We did not intend an influence on microbiological cultures but on the differential diagnosis with vasculitis and we have detailed appropriately in the revised manuscript. Line 150, 187, 271, 348 (and other occasions): we have corrected reporting always CNS instead of SNC. Line 163: we have changed to “Galactomann antigen on CSF was positive in 67% of cases.” Line 192-193: in the revised manuscript, we have simplified the sentence that was too long and confusing. Line 194: sorry for the mistake, we have corrected the spelling of Actinomyces spp. Line 211-214: we have rewritten the sentence, as requested. Line 216: “Can developed” has been changed to “can develop”. Line 220-226: in the revised manuscript we have elaborated on the source of Bacillus cereus in the study by Little et al. Line 243: we have corrected the sentence in the bracket. Line 246: Please elaborate on how your laboratory does 16S rRNA sequencing. 16S rRNA sequencing has been performed on fresh specimens with NGS. If NGS, you need to provide further information in terms of the results generated for respective organisms, including the number of reads, the NGS platform, etc.  We performed NGS (Ion Torrent, Thermofisher) read frequency 8000 for both pathogens. Line 254: we have changed to plasma. Line 266: we have changed in favour of “bacterial, fungal, and mycobacterial cultures”. Line 273-274: in the revised manuscript we have elaborated all abbreviations as requested. Line 280-281: One of the reasons for a negative 18s/28s sequencing is that the sequencing was performed on a specimen with previous exposure to anti-fungal. We agree with Reviewer’s comment, also the first specimen was done on antifungals (recent long exposure to voriconazole, some weeks of exposure to L-AMB) and unfortunately it was no more available for the additional analysis. Line 328-333: as suggested by the Reviewer, we have added and emphasisedthis point in the discussion section. Line 358: we have changed to “Treponema pallidum”. Line 363: we have corrected the mistake.

Figures: Line 389-390: “Timeline of brain MRI” has been deleted.

Reviewer 2 Report

Comments and Suggestions for Authors

The authors present a case of an immunocompromised patient with AML that had pulmonary invasive mold disease and later who developed CNS complications but despite all the investigations done no aetiology could be determined. The case highlights the need for a multidisciplinary approach and highlights the real life challenges experienced in the management of immunocompromised patient with an infective component. The case highlights the clinical reasoning that goes behind the use of anti microbial when even results present evidence. A very good clinical case with severe allergies important teaching points.

Since the review process of this paper takes time could the authors please report the final update on complete tapering of the steroids.

Comments on the Quality of English Language

A few English words may require editing such as more appropriate words may make manuscript easier to read e.g. line 88 patient was conducted to emergency room may read patient was rushed to emergency room or in-patient hospital admission may read on admission in hospital etc

Author Response

The authors present a case of an immunocompromised patient with AML that had pulmonary invasive mold disease and later who developed CNS complications but despite all the investigations done no aetiology could be determined. The case highlights the need for a multidisciplinary approach and highlights the real life challenges experienced in the management of immunocompromised patient with an infective component. The case highlights the clinical reasoning that goes behind the use of anti microbial when even results present evidence. A very good clinical case with severe allergies important teaching points.

Query 1: since the review process of this paper takes time could the authors please report the final update on complete tapering of the steroids.

Response 1: we thanks the reviewer for this comment and we have updated the clinical case to date in the revised manuscript.

Query 2: a few English words may require editing such as more appropriate words may make manuscript easier to read e.g. line 88 patient was conducted to emergency room may read patient was rushed to emergency room or in-patient hospital admission may read on admission in hospital etc.

Response 2: in the revised manuscript we have tried to improve the English with more appropriate words.

Round 2

Reviewer 1 Report

Comments and Suggestions for Authors

Thank you for the reply from the authors. They have addressed most of the comments raised. Here are some minor comments on the revised manuscript:

1. Line 88: The close bracket should be deleted in this line.

2. Lines 201 and 341: Names of the bacteria should be italicized.

3. Line 256: The close bracket is missing in this line.

4. Line 283: Enzyme instead of enzime

5. Line 289: "that resulted negative" should be changed to "and the result was negative"

6. Line 359: "was performed" should be changed to "was previously administered"

Author Response

Line 88: The close bracket was deleted

Lines 201 and 341: Names of the bacteria have been italicized.

Line 256: The close bracket was added.

Line 283:  enzime was replaced with enzyme

Line 289: "that resulted negative" was changed to "and the result was negative"

Line 359: "was performed" was changed to "was previously administered"